# Role of Neonatal Biomarkers of Exposure to Psychoactive Substances to Identify Maternal Socio-Demographic Determinants

**DOI:** 10.3390/biology10040296

**Published:** 2021-04-04

**Authors:** Pilar Jarque, Antonia Roca, Isabel Gomila, Emilia Marchei, Roberta Tittarelli, Miguel Ángel Elorza, Pilar Sanchís, Bernardino Barceló

**Affiliations:** 1Department of Pediatrics, Division of Neonatology, Son Espases University Hospital, Valldemossa Road, 79, 07120 Palma de Mallorca, Spain; pilar.jarque@ssib.es (P.J.); antonia.roca@ssib.es (A.R.); 2Pediatric Multidisciplinary Research Group, Balearic Islands Health Research Institute (IdISBa), Valldemossa Road, 79, 07120 Palma de Mallorca, Spain; 3Clinical Analysis Service, Son Llàtzer University Hospital, Manacor Road, 07198 Palma de Mallorca, Spain; isabel.gomila@hsll.es; 4Clinical Toxicology Research Group, Balearic Islands Health Research Institute (IdISBa), Valldemossa Road, 79, 07120 Palma de Mallorca, Spain; miguelangel.elorza@ssib.es; 5National Centre on Addiction and Doping, Istituto Superiore di Sanità, Viale Regina Elena, 299, 00161 Rome, Italy; emilia.marchei@iss.it; 6Department of Anatomical, Unit of Forensic Toxicology, Histological, Forensic and Orthopedic Sciences, Sapienza University of Rome, Piazzale Aldo Moro, 5, 00185 Rome, Italy; roberta.tittarelli@uniroma1.it; 7Clinical Analysis Service, Clinical Toxicology Unit, Son Espases University Hospital, Valldemossa Road, 79, 07120 Palma de Mallorca, Spain; 8Department of Chemistry, University of the Balearic Islands, Valldemossa Road, km 7.5, 07122 Palma de Mallorca, Spain; pilar.sanchis@uib.es; 9Research Group in Vascular and Metabolic Pathologies, Balearic Islands Health Research Institute (IdISBa), Valldemossa Road, 79, 07120 Palma de Mallorca, Spain

**Keywords:** biomarkers, prenatal exposure, psychoactive substances, meconium, sociodemographic factors, neonatal intensive care unit

## Abstract

**Simple Summary:**

The rapid identification of newborns exposure to psychoactive drugs allows an appropriate clinical care. This study tried to identify maternal profiles that help to identify newborns exposed to psychoactive drugs during pregnancy. Mothers were interviewed using a questionnaire. The biomarkers of fetal exposure were measured in meconium samples. Statistical analysis was performed to identify the maternal characteristics that were most likely to be associated with drug use during pregnancy. Of a total of 372 mothers, 49 (13.2%) tested positive for psychoactive drugs use: 24 (49.0%) for cannabis, 11 (22.5%) for ethyl glucuronide, six (12.2%) for cocaine, and eight (16.3%) for more than one psychoactive substance. The maternal characteristics that most likely identify substance use during pregnancy are: maternal age < 24 years, lack of pregnancy care, single-mother families, and active tobacco smoking. The profiles of prenatal maternal exposure identified in a clinical setting can be used to request specific detection tests for identifying newborns exposed to these drugs.

**Abstract:**

Background: The accurate assessment of fetal exposure to psychoactive substances provides the basis for appropriate clinical care of neonates. The objective of this study was to identify maternal socio-demographic profiles and risk factors for prenatal exposure to drugs of abuse by measuring biomarkers in neonatal matrices. Methods: A prospective, observational cohort study was completed. Biomarkers of fetal exposure were measured in meconium samples. The mothers were interviewed using a questionnaire. Univariate and multivariate logistic regression analyses were performed. Results: A total of 372 mothers were included, 49 (13.2%) testing positive for psychoactive substances use: 24 (49.0%) for cannabis, 11 (22.5%) for ethyl glucuronide, six (12.2%) for cocaine, and in eight (16.3%) more than one psychoactive substance. Mothers who consumed any psychoactive substance (29.7 ± 6.6 years) or cannabis (27.0 ± 5.7 years) were younger than non-users (32.8 ± 6.2 years, *p* < 0.05). Cocaine (50.0% vs. 96.9%, *p* < 0.05) and polydrug users (37.5% vs. 96.9%, *p* < 0.05) showed a lower levels of pregnancy care. Previous abortions were associated with the use of two or more psychoactive substances (87.5% vs. 37.8%, *p* < 0.05). Single-mother families (14.3% vs. 2.5%, *p* < 0.05) and mothers with primary level education (75.5% vs. 55.1%, *p* < 0.05) presented a higher consumption of psychoactive substances. Independent risk factors that are associated with prenatal exposure include: maternal age < 24 years (odds ratio: 2.56; 95% CI: 1.12–5.87), lack of pregnancy care (odds ratio: 7.27; 95%CI: 2.51–21.02), single-mother families (odds ratio: 4.98; 95%CI: 1.37–8.13), and active tobacco smoking (odds ratio: 8.13; 95%CI: 4.03–16.43). Conclusions: These results will allow us to develop several risk-based drug screening approaches to improve the early detection of exposed neonates.

## 1. Introduction

Psychoactive substances use during pregnancy is an important issue that can have significant and persistent adverse consequences for pregnant women and their infants. The impact of this use is a public health concern, and it has important implications for healthcare providers. Understanding trends, patterns of use, and outcomes is critical to developing prevention campaigns, building awareness, and providing effective care [1].

Infants that are exposed to drugs of abuse during pregnancy have a greater risk of entering a Unit Neonatal intensive care and neonatal intermediate therapy [2,3]. The accurate assessment of fetal exposure through the objective measurement of biomarkers allows for the identification of maternal socio-demographic determinants and the results have cascade effects on treatments and social services [4]. Thus, neonatal toxicology testing is emerging as a crucial service that hospital laboratories provide to both patient care teams and social services. Moreover, the measurement in nonconventional neonatal matrices, such as meconium and hair, provides a long historical record of prenatal exposure to certain drugs and it can account for different periods of gestation: meconium for the second and third trimester of gestation, fetal hair for the third [5].

In the DSM-5 (Diagnostic and Statistical Manual of Mental Disorders, 5th Edition), the diagnostic term ‘substance use disorders’ (SUDs) has been introduced. This term combines psychoactive substance abuse and dependence into one category, with a continuum of severity [6]. SUDs during pregnancy have health, social, and legal consequences. The prevalence of psychoactive substance use in pregnancy is uncertain, but it is assumed that, during the first trimester, it is comparable to that of the general population. At the end stages of pregnancy, there are some patients who continue to consume psychoactive substances [2]. Cannabis use prevalence ranged from 8.1% at the beginning of pregnancy to 2.5% in the last trimester [7]. The results from the 2013 National Survey on Drug Use and Health showed that illicit drug consumption was lower among pregnant women during the third trimester than during the first and second trimesters (2.4% vs. 9.0 and 4.8%). Similarly, alcohol use was lower during the second and third trimesters than during the first trimester (5.0 and 4.4% vs. 19.0%) [8].

Prenatal exposure to drugs of abuse can negatively affect the pregnancy itself and the development, growth, and maturation of the fetus. The clinical manifestations of this exposure range from abortion, intrauterine death, malformations, low weight, prematurity, fetal distress, premature rupture of membranes, asphyxia, and cerebral infarction to abnormal heart and breathing patterns. This prenatal exposure is also associated with the neonatal withdrawal syndrome and physical and neurobehavioral disorders that become evident in early childhood [2,3]. On the other hand, newborns that are exposed to maternal ethanol during pregnancy can develop a spectrum of physical, cognitive, and behavioral disabilities, known as fetal alcohol spectrum disorders (FASD), whose most severe form, including morphological abnormalities, is defined as fetal alcohol syndrome (FAS) [9,10,11].

Toxicological studies performed in Spain and Italy have documented that fetal exposure to drugs of abuse (heroin, cocaine, and cannabis) was 7.9–15.9% and exposure to ethanol was 1.7–45% [12,13,14,15,16]. The prevalence of alcohol, cannabis, hypnotherapy, cocaine, ecstasy, and amphetamine use in Mallorca in 2015 was 79.7, 15.3, 9.9, 5.7, 2.2, and 1.2%, respectively [17], but the prevalence of prenatal exposure to drugs of abuse in Mallorca is unknown. The island, with 896,038 inhabitants (2019 data from the Statistics Institute of the Balearic Islands), has 78% of the total population of the Balearic Islands. Mallorca is also a well-known international tourist destination.

National guidelines promote universal screening for substance use in pregnancy, but not universal testing [18]. Screening tests should be performed as a universally administered questionnaire designed to ascertain who is at high risk for having a substance use disorder in pregnancy [19]. Nevertheless, it is well known that maternal self-reports on psychoactive substance use history have proved to be unreliable [16]. 

In Spain, the Clinical Practice Guide for Care in Pregnancy and Puerperium suggests that pregnant women give up the consumption of psychoactive substances through cessation interventions. However, urine and alcohol drug screening is not given as a recommendation [20]. Similarly, The American College of Obstetricians and Gynecologists (ACOG) recommends routine screening for SUDs in all individuals, but screening is defined as questionnaires or patient interviews. The ACOG guideline states that the routine laboratory testing of biological specimens is not required [21].

There are several ethical, legal, and social considerations around toxicology tests in pregnancy that can be lost in the rush to test and confirm from a scientific and/or analytical perspective. The issue of consent is often raised in terms of drug screening, which in the specific case of pregnancy is complicated because drug use affects both the mother and neonate. Additionally, the ethical principal of “respect for persons” mandates that the woman give consent for the procedure given the social and legal ramifications of the test. Women also fear stigmatization and the legal consequences of drug use in pregnancy [22]. This leaves to individual institutions the decision to establish appropriate psychoactive substance testing policies.

The measurement of drugs and alcohol biomarkers in different neonatal matrices is an opportunity for clinical laboratories to assume a prominent role in the diagnostic management team for newborns and in the detection of SUDs in pregnant women [23]. Clinical laboratories can contribute to several distinct areas: the development of institutional and health system policies for psychoactive substances testing; education on the choice of specimen analysis; education that is related to specimen analysis; clinical interpretation of results; and, integration with maternal history and maternal toxicological results [24].

Despite the above considerations, the implementation of protocols for detecting prenatal psychoactive substances exposure through the analysis of biological biomarkers is scarce. Consequently, exposure cannot be detected during pregnancy, and it can only be diagnosed at birth if toxicological tests are performed. In the worst case, exposure will not be detected. This prevents early neonatal diagnosis and adequate medical treatment and social monitoring. Because newborns that are exposed to psychoactive substances during pregnancy have a higher risk of admission to the Neonatal Intensive and Intermediate Care Unit (NICU), this situation represents a challenge for their detection.

The measurement of biomarkers in neonatal matrices allows for the detection of this exposure. The accurate assessment of fetal exposure to psychoactive substances will provide correct information on the prevalence of drug abuse during pregnancy and consequent prenatal exposure to toxins in our NICU and will supply the basis for appropriate treatment and clinical and neurological follow-up of exposed newborns [2,13,23,25].

Therefore, the objective of this study was to identify maternal sociodemographic profiles and risk factors for prenatal exposure to psychoactive substance by measuring of biomarkers in neonatal matrices. The results will allow for the development of risk-based biomarker screening approaches. These approaches will enable the early detection of exposed neonates admitted to the NICU and rapid intervention.

## 2. Materials and Methods

### 2.1. Study Subjects

The work was a prospective observational study that was carried out in the NICU (Level IIIB) of the University Hospital Son Espases in Mallorca (Spain) between March 2018 and December 2019. Mother–infant dyads were recruited at the admission of neonates in the NICU. The following mother-infant dyads were excluded: mothers of neonates who presented with meconium aspiration syndrome, intrauterine intestinal perforation, delayed meconium evacuation, melaenic stools, severe asphyxia or death; and, mothers of neonates with no suspected prenatal exposure to psychoactive substances who were admitted on non-working days or during holidays periods or from whom meconium could not be collected.

### 2.2. Neonatal Biomarkers Testing

During the past decades, urine has been the specimen of choice for drugs of abuse screening. Neonatal urine are useful for determining acute exposure to drugs of abuse in the period immediately previous to delivery [23]. The two matrices of choice for neonatal research of biomarkers of exposure to alcohol in the last quarter or the last two trimesters of pregnancy are, respectively, the hair and meconium of the newborn.

Meconium is the first fecal matter passed by a neonate whose formation starts between the 12th and 16th week of gestation. Meconium analysis extends the window of detection of drug use to approximately the last 20 weeks of gestation, and it has been used to assess the prevalence of in utero drug exposure [23]. Neonatal hair starts growing during the last three to four months of pregnancy and therefore represents for exposure occurring in the last trimester [23].

Exposure was defined as the presence of psychoactive substances biomarkers in meconium samples that were collected within the first 24 h of life. The detection of drugs of abuse biomarkers was performed immediately and an aliquot was stored at −20 °C for the subsequent detection of Ethylglucuronide (EtG).

Only in neonates with suspected prenatal exposure and when there was no meconium sample available to perform drug analysis in this matrix, analysis was performed on urine and/or neonatal hair. Drug testing was performed in urine and neonatal hair in 12 cases: in four cases meconium sample was not available and in eight cases meconium amount was insufficient to confirmatory tests. Urine bags were used to collect neonatal urine samples while, for neonatal hair analysis, a sample of 100–200 mg of hair was collected.

#### 2.2.1. Meconium Toxicology Testing for Analysis of Illicit Drugs

Upon admission, meconium toxicology testing for analysis of illicit drugs biomarkers was performed using a combination of immunoassay and chromatography/mass spectrometry techniques. Meconium samples were homogenized and extracted, and the extracts were analyzed using an initial immunoassay (DRI^®^Assay, Abbott Laboratories Inc., North Chicago, IL, USA) for cocaine, opiates, cannabinoids (tetrahydrocannabinol: THC), amphetamines, ecstasy, and methadone, with cutoff concentrations of 20 ng/g for THC and 200 ng/g for the cocaine, opiate, amphetamine, ecstasy, and methadone metabolites [26,27]. Presumptive positive screens were confirmed using gas chromatography/mass spectrometry (GC/MS), with cutoffs at the limit of detection of the assays [27,28,29]: 50 ng/g for cocaine, opiate, amphetamine, ecstasy, and methadone metabolites; and, 5 ng/g for THC-COOH. The cocaine metabolites that were included in the GC/MS confirmation analysis were cocaine, benzoylecgonine (BE), and cocaethylene (CE); the opiate metabolites included morphine, codeine, and 6-monoacetylmorphine (6-MAM); the methadone metabolites included methadone and 2-ethylidene-1,5-dimethyl-3,3-diphenylpyrrolidine (EDDP); and, THC-COOH, amphetamine, methamphetamine, and 3,4-methylenedioxymethaphetamine (MDMA, known as ecstasy) were also confirmed by GC/MS.

#### 2.2.2. Meconium Toxicology Testing for Analysis of EtG

EtG was used as a biomarker for prenatal alcohol exposure. Meconium samples were analyzed using ultrahigh performance liquid chromatography coupled to tandem mass spectrometry (UPLC-MS/MS), with detection and quantification limits of 0.5 ng/g and 5 ng/g, respectively [30]. The cutoff point to consider EtG positive was 30 ng/g [31].

#### 2.2.3. Urine and Hair Toxicology Testing for Analysis of Illicit Drugs

Analysis of illicit drugs in urine were performed following the same protocol of meconium analysis [32,33,34]. The analysis of illicit drugs in hair was performed by GC-MS following previously published methods [32,35,36].

### 2.3. Data Collection

Upon admission of neonate to the NICU and after the mother gave informed consent, a maternal interview was conducted to determine sociodemographic information and past and current psychoactive substances use. The questionnaire was structured, based on a written script and previous reports, and conducted by a pediatrician [12,13,37].

Thirteen questions were included on sociodemographic characteristics (age, nationality, family type, sexual orientation, and educational level), obstetric history, pregnancy control, and psychoactive substance use habits during pregnancy. Pregnancy care was considered adequate if serological tests (HIV, syphilis, and HBV), combined first trimester screening, O’Sullivan’s test, and all three ultrasounds (first, second, and third trimesters) had been performed [20]. The substances investigated in this questionnaire included alcohol, cannabis, morphine, heroin, cocaine, amphetamines, ecstasy, methadone, alcohol, tobacco, and licit drugs used during pregnancy. Upon discharge, the cases that were referred to Social Services and the cases in which guardianship of the newborn was withdrawn were recorded.

The survey data and collected biological matrices (meconium, urine and hair) were coded in order to secure the participants’ privacy, and the local Human Research Ethics Committee (Research Ethics Committee of the Balearic Islands CEI-IB, project number IB 3538/17 IP) approved the study protocol.

### 2.4. Data Analyses

The data obtained from neonatal biomarkers testing and the mother’s interview were recorded in a Microsoft Office Excel 10 spreadsheet.

To obtain the demographic profile that is associated with prenatal exposure to psychoactive substances, the group with negative toxicological results for all of the tested substances was compared with the positive toxicological group for any psychoactive substance, the positive groups only for one psychoactive substance and the positive group for more than one psychoactive substance, respectively. In addition, the sociodemographic profiles associated with the consumption of cannabis, alcohol, or cocaine, were compared with each other to detect specific sociodemographic profiles that are associated with the use of cannabis, cocaine or alcohol. Values were expressed as the mean ± standard deviation, median [interquartile range] or frequency (percentage).

Comparisons between “positive” and “negative” toxicology groups were performed by independent t-test or Mann Whitney U for quantitative variables; and, chi-square or Fisher’s exact test for qualitative variables. Comparisons between “positive only for cannabis”, “positive only for alcohol”, “positive only for cocaine”, “positive for more than one drug”, and “negative toxicology” groups employed one-way ANOVA and LSD (Least Significant Difference) as a post-hoc test or Kruskal–Wallis test and Mann Whitney U test for continuous variables; and, chi-square for qualitative variables.

The Kappa coefficient was used to determine the agreement between the positive toxics detected in meconium and the consumption of positive toxics declared by the mother.

Univariate and multivariate binary logistic regression was used to identify the independent risk factors associated with prenatal exposure (positive toxicology), using as reference those cases with negative toxicology (odds ratio = 1). Initially, the sociodemographic variables with *p* < 0.05 were introduced into the model. The multivariate model was built step by step until the final model was obtained. The optimal cut-off values (those that optimize sensitivity and specificity) of the quantitative variables of the quantitative variables associated with psychoactive substance use were determined by receiver operating characteristic (ROC) curves and the maximum Youden (J) index, defined as sensitivity + specificity − 1. From the final model by binary logistic regression of the independent variables that are associated with psychoactive substance use, the probability for each case was calculated.

A two-tailed *p*-value less than 0.05 was considered to be statistically significant. Statistical analyses were performed using SPSS 27.0 (SPSS Inc., Chicago, IL, USA).

## 3. Results

During the 22-month recruitment period, 895 neonates were admitted to the NICU. Of these, 459 met the eligibility criteria and 372 (81.0%) consented to participate in the study. Of the 372 women included in the study, 229 (61.6%) were Spanish, with the mean (±standard deviation, SD) age being 32.4 ± 6.3 years; 53 (14.2%) had a higher university degree and 215 (57.8%) had primary education. Table 1 shows the sociodemographic profiles associated with prenatal exposure to psychoactive substances.

### 3.1. Neonatal Biomarkers Testing

The biomarkers testing results were negative in 323 (86.8%) cases and positive in 49 (13.2%) cases. Among the positive cases, 24 (49.0%) were for cannabis, 11 (22.5%) for EtG, six (12.2%) for cocaine, and in eight (16.3%) more than one psychoactive substance was detected.

The median concentration of EtG [quartile1-quartile3] in the meconium samples considered positive was: 71.1 [45.0–113.5] ng/g. 28 (7.5%) cases had EtG concentrations between 5 and 30 ng/g, with a median [interquartile range] of: 12.4 [9.4–18.1] ng/g.

Hair samples were positive for cocaine in eight cases, for cannabis in two cases and for cocaine and cannabis in one case. In eight cases, cocaine was confirmed in urine. One case was negative in both samples. EtG was detected in meconium samples in all cases. This means that consumption occurred during the second and third trimester of pregnancy.

When the positive toxicology results were compared with data recorded in the questionnaire, the agreement obtained for each substance were: 66.7% (*k* = 0.671; *p* < 0.001) for cannabis, 45.5% (*k* = 0.564; *p* < 0.001) for EtG, 16.7% (*k* = 0.260; *p* = 0.007) for cocaine, and 25.0% (*k* = 0.358; *p* = 0.001) for more than one psychoactive substance use.

### 3.2. Sociodemographic Profiles Associated with Prenatal Exposure to Psychoactive Substances

Table 1 reports the parental socioeconomic and demographic characteristics in relation to the toxicology results obtained by meconium biomarkers analysis. Only variables with statistically significant differences between groups are shown. The patterns were obtained by comparing the toxicology results with the sociodemographic data of the questionnaire.

#### 3.2.1. Age

Psychoactive substance use was associated with a significantly younger age of mothers when compared to non-users (29.7 ± 6.6 vs. 32.8 ± 6.2 years, *p* < 0.05). Cannabis use was associated with both younger mothers (27.0 ± 5.7 vs. 32.8 ± 6.2 years, *p* < 0.05) and younger fathers (30.9 ± 7.3 vs. 35.3 ± 6.7 years, *p* < 0.05). Moreover, the cannabis users were younger than the alcohol users (27.0 ± 5.7 vs. 33.3 ± 7.6 years, *p* < 0.05) and the cocaine users (27.0 ± 5.7 vs. 33.2 ± 6.5 years, *p* < 0.05). 

#### 3.2.2. Lack of Adequate Pregnancy Care

A lack of adequate pregnancy care was associated with substance use (77.6% vs. 96.9%, *p* < 0.05), specifically with cocaine (50.0% vs. 96.9%, *p* < 0.05) and more than one psychoactive substance (37.5% vs. 96.9%, *p* < 0.05). Cocaine users also showed lower levels of care than cannabis users (50.0% vs. 91.7%, *p* < 0.05).

#### 3.2.3. Previous Abortions

Exposure to more than one psychoactive substance during pregnancy was associated with a significantly higher percentage of previous abortions (87.5% vs. 37.8%, *p* < 0.05). 

#### 3.2.4. Single Mother

Single mothers used psychoactive substances (14.3% vs. 2.5%, *p* < 0.05), specifically cannabis (16.7% vs. 2.5%, *p* < 0.05) and more than one psychoactive substance (25.0% vs. 2.5%, *p* < 0.05), more frequently than mothers with a partner. 

#### 3.2.5. Academic Level

A lower education level of mothers was associated with substance use during pregnancy. More frequently, psychoactive substance-using mothers had only completed primary level education (75.5% vs. 55.1%, *p* < 0.05). In those with a university degree, psychoactive substance use was much lower (4.1% vs. 15.8%, *p* < 0.05).

The academic profiles were different between cannabis and alcohol consumers. Education only to primary level was more frequent in cannabis users than in alcohol users, both in mothers (91.7% vs. 36.4%, *p* < 0.05) and fathers (85.0% vs. 27.3%, *p* < 0.05). In contrast, holding a bachelor’s degree is less common in mothers and fathers who used cannabis as compared to those who consume alcohol (0% vs. 27.3%, *p* < 0.05 in both).

#### 3.2.6. Self-Reported Tobacco Smoking and Licit Drug Use

Self-reported tobacco smoking (59.2% vs. 14.9%, *p* < 0.05) and licit drug use (30.6% vs. 16.8%, *p* < 0.05) was significantly associated with psychoactive substance use. In addition, positive cases were significantly associated with tobacco consumption by the father (59.5% vs. 26.5%, *p* < 0.05).

#### 3.2.7. Referral to Social Services and Custody withdrawal

The reporting of cases to Social Services (63.3% vs. 1.9%, *p* < 0.05) and withdrawal of guardianship of the neonate (12.2% vs. 0.3%, *p* < 0.05) were more frequent in cases in which psychoactive substance use was detected. The notification to Social Services was frequent in cases of cannabis use (79.2% vs. 1.9%, *p* < 0.05) and cocaine use (83.3% vs. 1.9%, *p* < 0.05). There were no cases reported to Social Services relating to alcohol use.

### 3.3. Independent Sociodemographic Risk Factors Associated with Prenatal Exposure

The final model showed that the independent sociodemographic risk factors that were associated with prenatal exposure were: maternal age < 24 years (odds ratio: 2.56; 95%CI: 1.12–5.87), lack of pregnancy care (odds ratio: 7.27; 95%CI: 2.51–21.02), single mother (odds ratio: 4.98; 95%CI: 1.37–18.13), and mother with active tobacco smoking (odds ratio: 8.13; 95%CI: 4.03–16.43) (Figure 1).

## 4. Discussion

In this study, sociodemographic profiles and independent risk factors that were associated with prenatal exposure to psychoactive substance were identified by the measuring of biomarkers in neonatal matrices. To our knowledge, this is the first study of these characteristics carried out in our country in a NICU.

The prevalence estimates for prenatal substance use vary widely and they have been difficult to establish. The true prevalence of drug use among pregnant women is difficult to ascertain and widely vary across countries [38,39]. Therefore, clinical laboratories through the measurement of biomarkers of exposure to psychoactive substances in biological matrices can play an essential role in assessing the real fetal exposure, allowing for the generation of appropriate policies and interventions.

The top three substances detected, cannabis, EtG, and cocaine, showed a similar pattern to the drug abuse results that were published to Spain by the European Monitoring Centre for Drugs and Drug Addiction in women between aged 15 and 35 years, lending high reliability to the data [40]. As expected, cannabis was the most frequently detected substance. The incidence of prenatal cannabis use detected in this study (8.3%, including cases positives for more than one drug), was in the range of other studies performed using drug testing in meconium samples (2.8 to 10.3%) [21,41,42].

After cannabis, the second detected substance was EtG (4.3% of samples over established cut-off, including cases positives for more than one drug). Moreover, EtG was also detectable, although at levels that were below the established cutoff, in an additional 7.5% of patients. In agreement with previously reported data, our results show the high prevalence of women consuming a non-negligible amount of alcohol during their pregnancies [43].

The third detected substance was cocaine (3.2%, including cases positives for more than one drug). The obtained results were similar to the results reported in meconium samples in other studies, ranging from 1.2 to 5.6 [13,44,45].

Finally, in our study one sample was found positive to opiates and one positive to methadone in one mother in opioid substitution treatment. Ecstasy or amphetamines were not detected. These findings were lower than the results found in other cohorts of Mediterranean area [12,37,46].

The EtG levels found in the positive meconium samples were in the range of previous studies, with concentrations between 71-208 ng/mg [31,47,48]. Meconium samples were considered to be positive for cannabis and cocaine when they were above the detection limit of the assays (5 ng/g for THC-COOH and 50 ng/g for cocaine metabolites), but the levels were not quantified.

When considering the positive cases detected in our study (13.2%) and the annual average of live neonates (8053 infants in the last five years according to data from the Statistics Institute of the Balearic Islands), more than 1000 mothers and babies in Mallorca could test positive every year. However, this number is likely to be higher than the real one, since neonates that are exposed to psychoactive substances are more likely to be admitted to a NICU than those non-exposed. Nevertheless, this does help demonstrate the potential magnitude of this issue in the study area.

Drug use during pregnancy is a risk factor for both maternal and fetal complications. Additionally, children that are exposed to drug use are susceptible to many adverse long-term effects [2,3,4,10,49,50]. Some studies have associated cannabis use during pregnancy with lower birth weights and an increased incidence of tremors, exaggerated startles, and diminished crying in the neonate and longer gestations. In addition, children exposed to cannabis had a lower performance on standardized tests, indicating that long-term behavioral and neurodevelopmental issues may occur in these children [1,50].

With cannabis being touted on the Internet as a safe treatment of nausea during pregnancy, current rates of use of cannabis during pregnancy are a concern. However, there are currently no indications for its use during pregnancy and research is needed before cannabis can be considered for use in hyperemesis gravidarum. Koren et al. suggested that cannabis should be tested in appropriately powered control trials for this severe and protracted maternal condition, addressing both maternal effect and potential adverse fetal effects [51,52]. Pregnant women who are using cannabis in pregnancy should be counseled about the lack of safety data and the possible adverse effects of THC on the developing fetus and referred to their health care provider for alternative treatments that have better pregnancy-specific safety data [40,50].

Alcohol exposure during pregnancy can have many serious consequences for the offspring, including miscarriage, stillbirth, preterm birth, intrauterine growth retardation, teratogenicity, and alcohol-related neurodevelopmental disorders. The fetal alcohol spectrum disorder (FASD) describes the consequences that are associated with prenatal alcohol exposure. The clinical presentation of the disease is inconsistent, some lacking evidence of central nervous system neurodevelopment abnormalities. Typically, for the definite diagnosis of FASD, the confirmation of prenatal ethanol exposure is needed, because some of the characteristics of the disease, in particular, physical, may be absent [22,53,54].

Prenatal cocaine exposure is associated with low birth weight, prematurity, spontaneous abortions, stillbirths, and microcephaly. In addition, maternal hypertension, tachycardia, vasoconstriction, increased uterine contractility, and placenta complications have been described, including placental abruption, and an increased risk of diminished blood flow to the fetus. Exposure to cocaine during childhood can increase the risk for hypertension, ventricular arrhythmia, seizures, and intracranial bleeding. Behavioral problems also present themselves during development to both prenatally and postnatally exposed children. Infants that are exposed in utero are found to have attention deficits with an increased incidence of attention-deficit/hyperactivity disorder and transient central and autonomic nervous system signs and symptoms. Prenatal cocaine exposure is associated to sudden infant death syndrome [54,55,56,57].

Poly-substance use is common among substance users, and this and the acuteness or the chronicity of the use may also modify or exacerbate the individual effects on maternal and fetal health [1]. In our study, poly-substance use was associated significantly with previous abortions; these findings were in accordance with other studies [12,13,37,41,46,58]. As in other studies, SUDs were not associated with a higher frequency of preterm births [12,13,14,37,46].

Moreover, pregnant women who are substance users often have complex social and mental health issues and these women need access to assertive outreach care from specialists in addiction and mental health. There is also the additional risk of transmission of viral infections (hepatitis B, hepatitis C, or HIV).

Taken together, these adverse consequences of substance use to the developing fetus and mother are complex, and efforts should be made to identify and support women who are using substances. It is important that health care providers understand the risk factors and adverse consequences of substance use during pregnancy on maternal and fetal health to provide appropriate advice to substance-using pregnant women [1].

Th obtained data underline the usefulness of meconium testing as an analytical tool for a more accurate identification of neonates exposed in utero to drugs of abuse compared to the identification based only on maternal questionnaire. The toxicological results objectively demonstrated that sociodemographic characteristics, such as maternal age, lack of pregnancy care, being a single mother, or smoking increased the risk of prenatal exposure to toxic substances in the neonate. Our findings are in agreement with the well-known under-reporting of alcohol and drug use by pregnant women [59].

The average age of mothers that were included in our study (32.4 years) was similar to that previously reported in the international literature [37,42,58,60]. Psychoactive substance use was significantly associated with younger age, as reported in other studies [42]. Pregnant cannabis users, as well as her partners, were significantly younger than non-users and cocaine and alcohol users. The pregnant cannabis users were younger than those that were reported in other studies (28.9–30.1 years) [5,13,37], but pregnant cocaine and alcohol users were older [12,13,46,48]. The results showed that mothers under 24 were over three and a half times more likely to use psychoactive substances during pregnancy. Thus, there is a need to target initiatives to prevent prenatal drug exposure, especially cannabis, at younger age groups.

Many studies have documented that substance-using women, particularly cocaine or opiate users, are significantly less likely to obtain pregnancy care, despite accruing greater benefit from it [14,40,52,61,62]. Our regression results showed that mothers with a lack of pregnancy care was over eight times more likely to use psychoactive substances. Moreover, this lack of pregnancy care was significantly associated with the use of cocaine or of more than one psychoactive substance. The low rate of pregnancy care in cocaine and poly-drug users could be explained by: (1) surrounding circumstances and chaotic lifestyles, (2) resource or attitudinal barriers to care, such as fear of police report, or (3) some direct, disruptive effect of the particular substance [2,61].

In contrast, the lack of pregnancy care was not associated with cannabis or alcohol use. Schempf el al. also found cannabis to be not independently related to pregnancy care [61].

The adequate pregnancy care that is associated with alcohol use could be explained by the underestimation of alcohol consumption by the mothers themselves [48], which implies adequate pregnancy care, despite continuing with substance use.

Co-use of tobacco and illicit drugs, particularly cannabis, is relatively high during pregnancy, as reported previously [47,52]. The results indicate that women who smoke cigarettes are more than six and a half times more likely use psychoactive substances during pregnancy. Given the strong association between smoking and other drug use, clinicians should routinely assess for illicit drug use in women who smoke during pregnancy [63]. Smoking tobacco has a significant relationship with cannabis use during pregnancy, but not with cocaine use, as also reported in other studies [5,13,37,42,46]. As in our work, tobacco consumption was declared by the mother and no toxicological analyses were performed.

Our results showed a significantly relation between licit drugs and illicit drug use in pregnant women. Published data on this association are scarce, but prescription drug abuse, especially opiates, benzodiazepines, and stimulants prescribed for attention-deficit/hyperactivity disorder, often co-occurs with other illicit substances or alcohol abuse [61]. Joya et al. [46] founded a correlation between cocaine and antidepressant use. The association between licit and illicit drug represents an increased risk to the mother and the fetus.

Even exposure to drugs of abuse have been associated with lack of education, some studies found no associations with academic level [37,46]. Our data showed that only having a primary level education was more common in cannabis users, both in mothers and fathers alike, in agreement with other studies [64,65]. The association between alcohol consumption and a higher academic level has been described previously [60]. In our study, an association was found between fathers with a bachelor’s degree and a higher frequency of alcohol consumption, although no differences were found in mothers. No clear association was found between academic level and cocaine use.

In a previous study that was performed in Spain, of the mothers who tested positive for psychoactive substance use, 40–60% had custody of their child removed [66]. In our population, child removed frequencies were lower, with a maximum of 25% in cases of polydrug use.

The referral to Social Services of cases of cannabis, cocaine and other psychoactive substance use was very frequent. However, the results of alcohol exposure were not available in the perinatal period and detected cases of alcohol consumption could not be referred to Social Services. These data suggest that social services may need to be involved with a larger proportion of mothers as, there may be mothers with addiction issues needing support, especially related to non-detected alcohol consumption.

Our results confirmed previous data in the literature, with psychoactive substance use during pregnancy being independently associated with low maternal age, a lack of pregnancy care, single mother status, and smoking habits [13,66,67,68,69]. These factors can lead to suspicion of a SUD, and can be used for the purpose of preventive health and policy strategies, especially in cases when the mother’s lack of awareness regarding health care, pregnancy planning, and low economic resources are also evident.

In our study, the potential SUDs in mothers were identified by the neonatologists. This was done due to the lack of attention to pregnancy and detection of psychoactive substance use, which meant that SUDs were not detected during pregnancy. Therefore, these studies reinforce the need to strengthen strategies for identifying such women.

Other risk factors that must be assessed in the preventive health and policy strategies are a history of mental illness [4] and domestic violence, as well as the presence of infectious diseases and employment status of the mother. These factors were outside the scope of the current study.

The limitations of this study first include that biomarkers of tobacco, new synthetic psychoactive drugs, or licit drug use were not analyzed. Mothers are often reluctant to admit having used drugs or they may not even be aware that they had been using a medication, so the measurement of drug concentrations (licit or illicit drugs) provides useful information for immediate infant treatment and subsequent medical follow-up.

Secondly, the results of alcohol exposure were not available in the perinatal period, which made it impossible to use them in early clinical or social health management. Finally, biomarkers analysis in meconium samples only detects psychoactive substance use in the second and third trimesters of pregnancy.

## 5. Conclusions

In summary, the risk factors that were identified in a clinical setting can be used to improve the detection of prenatal exposure to psychoactive substances. Our study demonstrates the usefulness of these factors in a real scenario: how the NICU and clinical toxicology laboratory work in real time for the early detection of exposed neonates.

Furthermore, the study showed that cannabis is the most frequently detected psychoactive substance in neonates admitted to our NICU, followed by alcohol and cocaine, reinforcing the need of suitable biomarkers for their detection. These results will allow us to develop several risk-based drug screening approaches to improve the early detection of exposed neonates admitted to the NICU and provide rapid intervention. The laboratory results have cascading impacts on effective medical care and treatment if this service works in real time.

The prevalence estimates for prenatal substance use vary widely and have been difficult to establish. Therefore, the work of clinical laboratory, as presented in this study, through the measurement of biomarkers of exposure in biological matrices play an essential role in assessing the real fetal exposure, allowing for the generation of appropriate policies and interventions.

The data can be used for the purpose of preventive health and policy strategies that aimed to avoid and detect prenatal exposure to drugs of abuse. Additionally, there is a need to target initiatives at different social groups to help prevent drug use during pregnancy and provide support, keeping in mind that it is important to consider that substance use occurs within a complex context.

## Figures and Tables

**Figure 1 biology-10-00296-f001:**
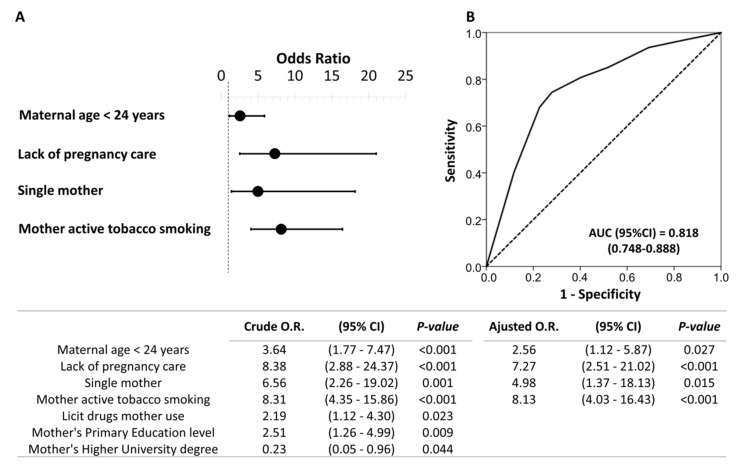
Binary logistic regression of the sociodemographic risk variables associated with prenatal psychoactive substance exposure. Forest plot (**A**) and receiver operating characteristic (ROC) curve (**B**) of the multivariate model obtained. Note: The table indicates the crude and adjusted odds ratios (O.R.) with their confidence intervals in parentheses. The optimal cutoff values were determined by the maximum Youden (J) index, defined as sensitivity + specificity − 1. A comparison of the expected and observed frequencies was made by the Hosmer–Lemeshow goodness-of-fit test (*p* = 0.551) and by its ROC curve and its area under the curve (AUC) indicating a good fit for the model.

**Table 1 biology-10-00296-t001:** Sociodemographic profiles associated with prenatal exposure to psychoactive substances. Only variables with statistically significant differences between groups are shown.

Variable [*n* (%)]	All Cases (*n* = 372)	Negative Toxicology (*n* = 323)	Positive Toxicology (*n* = 49)	Positive Only for Cannabis (*n* = 24)	Positive Only for EtG (*n* = 11)	Positive Only for Cocaine (*n* = 6)	Positive for More Than One Drug *(n* = 8)
Maternal age (years, mean ± SD)	32.4 ± 6.3	32.8 ± 6.2	29.7 ± 6.6 *	27.0 ± 5.7 *^ab^	33.3 ± 7.6	33.2 ± 6.5	30.0 ± 5.0
Paternal age (years, mean ± SD)	35.1 ± 6.8	35.3 ± 6.7	33.7 ± 7.2	30.9 ± 7.3 *^b^	35.8 ± 8.1	39.0 ± 4.4	33.8 ± 3.3
Adequate pregnancy care	351 (94.4)	313 (96.9)	38 (77.6) *	22 (91.7) ^b^	10 (90.9)	3 (50.0) *	3 (37.5) *
Previous abortions	146 (39.2)	122 (37.8)	24 (49.0)	7 (29.2)	6 (54.5)	4 (66.7)	7 (87.5) *
Self-reported mother’s use							
Tobacco	77 (20.7)	48 (14.9)	29 (59.2) *	16 (66.7) *	4 (36.4)	2 (33.3)	7 (87.5) *
Licit drugs	69 (18.5)	54 (16.8)	15 (30.6) *	5 (20.8)	4 (36.4)	1 (16.7)	5 (62.5) *
Self-reported father’s use							
Tobacco	108 (29.0)	83 (26.5)	25 (59.5) *	16 (80.0) *	2 (18.2)	3 (60.0)	4 (66.7) *
Single mother	15 (4.0)	8 (2.5)	7 (14.3) *	4 (16.7) *	0 (0)	1 (16.7)	2 (25.0) *
Mother’s academic level							
Primary education	215 (57.8)	178 (55.1)	37 (75.5) *	22 (91.7) *^a^	4 (36.4)	5 (83.3)	6 (75.0)
Vocational education and training	52 (14.0)	46 (14.2)	6 (12.2)	2 (8.3)	2 (18.2)	0 (0)	2 (25.0)
Bachelor’s degree	52 (14.0)	48 (14.9)	4 (8.2)	0 (0.0) ^a^	3 (27.3)	1 (16.7)	0 (0)
Higher university degree	53 (14.2)	51 (15.8)	2 (4.1) *	0 (0.0) *	2 (18.2)	0 (0)	0 (0)
Father’s academic level							
Primary education	210 (56.5)	182 (57.8)	28 (66.7)	17 (85.0) *^a^	3 (27.3)	3 (60.0)	5 (83.3)
Vocational education and training	93 (25.0)	84 (26.7)	9 (21.4)	3 (15.0)	4 (36.4)	1 (20.0)	1 (16.7)
Bachelor’s degree	27 (7.3)	23 (7.3)	4 (9.5)	0 (0) ^a^	3 (27.3) *	1 (20.0)	0 (0)
Higher university degree	30 (8.1)	29 (9.2)	1 (2.4)	0 (0)	1 (9.1)	0 (0)	0 (0)
Referral to Social Services	37 (9.9)	6 (1.9)	31 (63.3) *	19 (79.2) *^a^	0 (0) ^c^	5 (83.3) *	7 (87.5) *
Custody withdrawal	7 (1.9)	1 (0.3)	6 (12.2) *	3 (12.5) *	0 (0)	1 (16.7) *	2 (25.0) *

** p* < 0.05: statistically significant differences between negative and positive toxicology groups. ^a^
*p* < 0.05: statistically significant differences between cannabis and EtG groups. ^b^
*p* < 0.05: statistically significant differences between cannabis and cocaine groups. ^c^
*p* < 0.05: statistically significant differences between EtG and cocaine groups. EtG: ethyl glucuronide; SD: Standard deviation.

## Data Availability

The data presented in this study will be made available upon request to the corresponding authors.

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
