# Peer review of "Role of Neonatal Biomarkers of Exposure to Psychoactive Substances to Identify Maternal Socio-Demographic Determinants"

_biology, 2021, doi:10.3390/biology10040296_

Round 1
Reviewer 1 Report
My previous comments have been addressed
Author Response
Reviewer 1 has accepted all the previos point-by-point response.
Reviewer 2 Report
General
The authors have addressed a lot of my comments from my initial review. However I feel that the revised discussion doesn’t sufficiently address the issues that I raised. The formatting of the document needs to be checked to ensure a consistent approach to paragraph use and indenting. I have made the following comments to facilitate a further revision:
- Introduction
Page 2 (apologies line numbers missing on my manuscript)
Make a paragraph out of the first three paragraphs. Also make a single paragraph out of para 3 and 5
- Materials and Methods
2.1 Page 4 (apologies line numbers missing on my manuscript)
Could you add a short explanation- e.g. “due to difficulties obtaining adequate meconium samples the following mother-infant dyads were excluded”
2.2 Page 4 (apologies line numbers missing on my manuscript)
Para 4- Were the urine and hair samples analysed separately or combined with the meconium results
Page 5 (apologies line numbers missing on my manuscript)
I think the layout of the information in the first two paragraphs could be improved. Throughout the paper I am getting confused in terms of what represents toxicology and what represents questionnaire results. If this paragraph was reformatted I think it would help. For example if each drug was a sub heading with the test undertaken in text underneath it would help. At the moment the test is described and then the drug tested for comes as secondary and doesn’t stand out.
2.3 Page 5 (apologies line numbers missing on my manuscript)
Insert ‘previous’ instead of ‘previously’
2.4 Page 5 (apologies line numbers missing on my manuscript)
The data analysis section should contain information about the correlational analysis. What type of analysis was undertaken, and what type of correlation was employed?
- Results
P6- Join up first two paragraphs
3.1 Page 6 (apologies line numbers missing on my manuscript)
The correlation results are too summarised and I do not understand how the summary results were derived. Correlations for each substance should have been undertaken and statistical significance should have been calculated for each and presented in brackets after text explanation.
3.2 Page 6 apologies line numbers missing on my manuscript)
First line- ‘in relation to ‘toxicology results”
3.2 Page 9 line 11-43
This is much more use friendly. The significance levels do need to be put in brackets after the statement of association is given. It may also be useful to use sub headings to improve it further- e.g. Age, lack of pregnancy care etc. Lack of pregnancy care should be defined somewhere in the paper. I do not see it.
I am assuming that these patterns are comparing toxicology results with sociodemographic data on the questionnaire. This should be clarified.
- Discussion p10, but it prints p2
Overall the discussion needs to be reworked. It should be structured better, with the key factors influencing substance use given first. These should be discussed, reinforcing where appropriate with the regression results, reference to supporting literature, and policy implications as appropriate.
Page 10 line 58
The overall levels need to be put in here and discussed in terms of comparable literature (using meconium if possible) to determine the extent of the issue. The top 3 substance should be given and the significance in terms of the potential health impacts of use should be stated, giving supporting literature as appropriate. Also are there any implications in terms of the care needed for mothers and their babies. What level of risk do the levels found have for mothers and babies? Based on the 13.2% figure, perhaps the authors could estimate how many mothers and babies in Mallorca test positive every year. This may help to reinforce the point about needing to address their care needs.
Page 10 line 59-63
This needs slight rephrasing as it doesn’t read well. This point should be strengthened by linking to the correlation results.
Page 10 line 64-69
The regression results should be added here to reinforce the importance of age. For example, those under 24 are over three and a half times more likely to use psychoactive substances during pregnancy. It should be stated that there is a need to target initiatives at younger age groups.
Page 10 line 71-page 11 (printed as 3) line 78
Lack of pregnancy care is the most influential factor and this point should be emphasised. This can be facilitated by reinforcing with the regression results which shows that those with a lack of pregnancy care are over eight times more likely to use psychoactive substances. The importance of pregnancy/prenatal care should also be emphasised with supporting literature. I am not clear what lack of pregnancy care refers to and this should be made explicit. Terminology needs to be standardised as well. For example the term lack of prenatal care is used interchangeably. These findings are important and I appreciate the authors have discussed them, but I feel they require further discussion. For example, it appears that drug users may not seek prenatal care for fear of drug detection. Are there not policy and health implications here.
Page 11 (printed as 3) line 80-82
The results for smoking are not discussed. This sections reads more like results. Tobacco smoking is the second most important factor in the regression results and the findings require discussion.
Page 11 (printed as 3) line 84-87
This seems to be placed in the discussion without context. This should either be removed or kinked to a study finding and discussion point being made.
Page 11 (printed as 3) line 88-90
The issue of polydrug use and abortions dues not fit here. It should be placed at the start of the discussion where the overall patterns are given to support the point about the health implications of psychoactive drug use in pregnancy.
Page 11 (printed as 3) line 92-94
This should be removed from the finding as this was not statistically significant in the current study.
Page 11 (printed as 3) line 95-101
Mothers lack of education is a key factor in this study which should be reflected in the discussion. The current discussion lacks focus in terms of the key issues and does not talk about the lack of education or give any policy implications. Are these early school leavers and what age group does it refer to?
Page 11 (printed as 3) line 103-111
The authors raise important issues here. However this paragraph doesn’t read well and needs to be reworded to improve the flow of the argument.
Page 11 (printed as 3) line 119-122
Authors need to state that these were outside the scope of the current study. Please clarify educational level is here as this was included in the study and is dealt with earlier in the discussion.
Page 11 (printed as 3) line 123-126
This needs to be rephrased and the point needs to be integrated better into the discussion, linking better with the findings.
Page 12 (printed as 4) line 132-135
Could the questionnaires not have been used to identify people at risk in terms of alcohol. Surely this could be discussed.
Conclusions
Page 12 (printed as 4) line 137-138
There needs to be a much stronger concluding statement. To conclude that the study reported what had been found in previous studies is not a good way to conclude. This issue was highlighted in my previous comments, but has been left unchanged by the authors.
Page 12 (printed as 4) line 147-152
This point should also be made at the start of the discussion and repeated here in more summarised format at the start of the conclusion as this is a strong point.
Page 12 (printed as 4) line 156-158
“Therefore…professionals”- this needs strengthening as it doesn’t appear to explicitly link in with the findings
Author Response
I attach point-by-point response to the reviewer two.

Round 2
Reviewer 2 Report
Comments to authors: Role of neonatal biomarkers of exposure to psychoactive substances to identify maternal sociodemographic determinants
General
I can see the authors have put a lot of work into addressing my comments. The paper is now much stronger, and provides useful information to facilite policy and decision making in this important topic area. I do hae some further edits/suggestions. This are minor but need to be changed to help clarify some of the points being made. I am confident that these can be quickly rectified by the authors.
- Materials and Methods
2.1 Page 4 (apologies line numbers missing on my manuscript)
I do not see the specific change made by the authors, but I am nevertheless happy with the paragraph
2.23-2.24, Page 5 (apologies line numbers missing on my manuscript)
Merge urine and hair toxicology testing together into 2.23 which I feel will help the flow of paper.
2.3 Page 5 line 1 (apologies line numbers missing on my manuscript)
Upon admitting…
2.4 Page 5
Join line 1 and 2 to make one paragraph. Also add spct to “to psychoactive” in line 2.
2.4 Page 6
I am happy with the content overall. Could the authors review the use of paragraphs which currently is not user friendly.
In terms of the correlation analysis, I do not see where that is presented in the results. If it is not used in the results that are presented, it needs to be removed. The only reference I saw to correlation in the results was actually the kappa test results.
- Results
3.1 Page 6 last para- page 7 first para (apologies line numbers missing on my manuscript)
These don’t appear to me to be correlation results. These are kappa results. I found this confusiong. The text needs to be changed to remove any ambiguity. If this represents kappa findings then the term “correlation” should be removed and it should be made explicit that kappa was used.to avoid confusion. I am happy with the use of the kappa statistic.
3.2.6 Page 1 line 37
Insert space- “licit drug use”
3.2.7 Page 1 line 49-51
I don’t think this adds much to the paper and suggest that it should be removed to help maintain focus and structure.
3.3 Page 2 line 61-66
This long figure title is not user friendly. Please make a shorter figure title with the bulk of the explanatory text inserted as a sepeate footnote.
- Discussion
Page 3 line 97 and 99
Replace founded with found.
Page 3 line 109
Add a sentence highlighting implications such as “Nevertheless, this does help demonstrate the potential magnitude of this issue in the study area.
Page 3 line 116
Insert text on cannabis from later in discussion (see later comment)
Page 4 line 139
Can the authors clarify that correlational analysis was undertaken here. The nature of the variables compared suggests that this test was not undertaken. If correlations were not undertaken could the authors replace correlation with “association”
Page 4 line 143
Replace addictions with addiction.
Page 4 line 153-156 “In our study….women”
This paragraph doesn’t seem to fit in here. Could the authors check to see if it would be best placed somewhere else in the discussion. It would also need rephrasing as the point the authors are trying to make is not clear.
Page 4 line 184 page 5 line 192
This section needs to be reworked. For example it appears from a quick look myself that there are journal papers that support and do not support the use of cannabis for treatment of nausea. (e.g. 1. Koren, G., Cohen, R. The use of cannabis for Hyperemesis Gravidarum (HG). J Cannabis Res 2, 4 (2020). https://doi.org/10.1186/s42238-020-0017-6). 2. Roberson EK, Patrick WK, Hurwitz EL. Marijuana use and maternal experiences of severe nausea during pregnancy in Hawai'i. Hawaii J Med Public Health. 2014;73(9):283-287. As such a stronger argument needs to be given to support the need to advise pregnanat mothers not to use cannabis.
Page 5 line 193-195
This paragraph doesn’t seem to fit here and may be better placed elsewhere; perhaps combined with another section that deals with alcohol- for example the para ending at page 3 line 90. In addition this paragraph would need to be reworded as it is not clear to me what message the authors are trying to convey.
Page 5 line 205
“a significant relationship”
Page 5 line 220-222
This needs to be reworded as it is unclear which represents the study findings and which represents the literature.
Page 5 line 237
Remove nevertheless
Page 6 line 242
“provides” useful
Page 6 line 252
Reword this line.
Conclusions
Page 12 (printed as 4) line 137-138
There needs to be a much stronger concluding statement. To conclude that the study reported what had been found in previous studies is not a good way to conclude. This issue was highlighted in my previous comments, but has been left unchanged by the authors.
Page 12 (printed as 4) line 147-152
This point should also be made at the start of the discussion and repeated here in more summarised format at the start of the conclusion as this is a strong point.
Page 12 (printed as 4) line 156-158
“Therefore…professionals”- this needs strengthening as it doesn’t appear to explicitly link in with the findings
Author Response
See attached doc.

This manuscript is a resubmission of an earlier submission. The following is a list of the peer review reports and author responses from that submission.
Round 1
Reviewer 1 Report
The manuscript is well written and presents interesting findings, which are nicely discussed.
I have two main comments:
- Why NPS were not screened for? Are NPS common in the author’s country? Please discuss
- At page 4, lines 141-142, the authors declare that also urine and hair were collected. How were these samples collected, especially urine? How many cases? What about the results? Can the authors provide a breakdown of the results? The meaning of positive in meconium, hair and urine is clearly very different, also in terms of mothers’ use of substances. For example, EtG was detected in 11 cases, but in which specimen?
Author Response
The manuscript is well written and presents interesting findings, which are nicely discussed.
I have two main comments:
- Why NPS were not screened for? Are NPS common in the author’s country? Please discuss
AMENDED. There is currently little to no data available on perinatal toxicity of NPS. The demographics of prenatal exposure to NPS are largely unknown in the author’s country. In none of the cases included in the study was any suspicion of prenatal exposure to NPS. Documenting the use of NPS is challenging due to low drug concentrations and a lack of detection with the usual toxicological screening methods.
- At page 4, lines 141-142, the authors declare that also urine and hair were collected.
How were these samples collected, especially urine?
AMENDED. Urine collection bags were used to collect neonatal urine samples. For neonatal hair analysis, a sample of 100-200 mg of hair was collected.
How many cases?
AMENDED. Drug testing was performed in urine and neonatal hair in 12 cases, when a meconium sample was not available (4 cases) or was insufficient to confirmatory tests (8 cases). All these cases were in cases of suspected prenatal exposure.
What about the results? Can the authors provide a breakdown of the results?
AMENDED. Hair tests were positive for cocaine in 8 cases, for cannabis in 2 cases and for cocaine and cannabis in one case. In 8 cases, cocaine was confirmed in urine. One case was negative in both samples
The meaning of positive in meconium, hair and urine is clearly very different, also in terms of mothers’ use of substances. For example, EtG was detected in 11 cases, but in which specimen?
AMENDED. EtG was detected in meconium samples in all cases. This means that consumption occurred during the second and third trimester of pregnancy.

Reviewer 2 Report
Comments to authors: Role of neonatal biomarkers of exposure to psychoactive substances to identify maternal sociodemographic determinants
General
This is an important topic area and has important implications in terms of the future delivery of care to mothers and their new-borns, and also in terms of the use of toxicology results from meconium samples. However it does require significant improvements in terms of structure and focus, to ensure that key issues and policy implications can be determined. I hope the following detailed comments will facilitate the author(s) in revising the manuscript:
Simple summary
Page 1 Line 35
Put percent in brackets after 49.
Abstract
Page 2 Line 47
Put percent in brackets after 49.
Page 2 Line 49-53
Put significance levels in for these results.
Introduction
Page 2 line 63
A short overview of the key consequences for pregnant women and infants should be provided.
Page 2 line 68
“Allows identity maternal maternal,.,.”- suggest replace with “allows the identification of….”
Page 2 line 71-72
“of drug detection window”- this sentence does not flow correctly and requires minor rewording.
Page 2 line 83
It would be useful to highlight what percentage continue to use from other studies. In addiction it would be useful to explain the implications here of continued use during pregnancy.
Page 2 line 85
‘drugs of abuse’- there needs to be clarity in terms of terminology throughout the paper. It should not be assumed that the reader knows what these drugs are. It is not clear whether the studies quoted refer to the general population or pregnant women, or neonatal infants etc
Page 2 line 86-89
IT would be useful to provide background context in terms of SUD in Mallorca
Page 2 line 92
Expand stating why toxicology testing in pregnant women usually not recommended.
Page 3 line 90-99
It might be worth starting with the Spanish position, and then state that similar positions are employed in other countries, with an absence use of toxicology data (giving reference to the American guidelines and also some from other countries.
Page 3 line 98
‘in the hands of’- this should be rephrased or put in single inverted commas.
Page 3 line 101
Delete “In this scenario”
Page 3 line 124
Biomarker
Materials and Methods
Page 3 line 134-136
I don’t understand why these were excluded. This exclusion requires clarification.
Page 3 line 138
Short overview of meconium and rationale for use to analyse exposure, compared for example to hair or urine.
Page 4 line 141
What does EtG stand for?
Page 4 line 141-142
This is poorly explained. Why were these tests undertaken, and do they form part of the analysis in this paper?
Page 4 line 167
Further detail required. Was this before or after the baby was born? Was the written script a validated instrument, had it been piloted?
Page 4 line 174
Due to the fact that questions were asked about illegal substance use, I feel it would have been more appropriate to have used a self administered questionnaire. It would have been useful to correlate toxicology results with questions about psychoactive substances. The results appear to only show toxicology results. Is this correct and if this is the case, why were self reports collected?
Page 4 line 179-189
Further information in terms of how the survey and toxicology data was matched should be given. Were procedures put in place to conform to GDPR. Were there any ethical issues that had to be addressed? Was it explained to participants that their survey data was to be matched with toxicology results.? Was permission required to undertake the toxicology tests? Further detail required which would benefit others considering similar research.
Page 4 line 187
I am not sure what was done here. Please explain/
Page 5 line 198
? 1.
Results
Page 5 line 203-206
A summary of the sociodemographic profile of participants should be provided.
Page 5 line 209
(13.2%)
Page 5 line 211-216
Unless the authors provide justification, I feel that due to the small numbers involved, that this analysis is not required. However if the authors feel that this is important, then there will be a need to ensure the issues raised are included in the discussion
Page 6-9 tables 1-2.
The tables are large and not particularly user friendly. The authors should revisit to determine whether there is a better way to present findings. For example consideration should be given to presenting the core findings as opposed to detailed disaggregation of a small sample.
Page 10 line 234-253
These results are presented in a laborious and repetitive way. The key results should be presented in the text in a summarised user friendly format, with detailed results provided in table format. Currently too many patterns and relationships are presented which can mask the key patterns emerging. In addition, the numbers involved for some of the patterns are small, and I would query whether the size is sufficient to permit such analysis.
Page 10 line 279
How is “lack of pregnancy care” defined.
Discussion
Overall the discussion lacks focus. It presents lots of relationships that were found and then cites similar findings from the literature. Whilst comparisons withother studies is generally useful, in this case it fails to provide any real impact in terms of core issues of concern. There are key findings in the study, but these are lost in the patterns and relationships that are reported. I feel that a more focused discussion would have been more appropriate, perhaps structuring around the core drivers of the multiple logistical regression output. It would be better to take each of these, compare to other studies and discuss the implications. The utility of using meconium for toxicology results also warrants discussion.
Page 11 line 297
NICU?
Page 11 line 301-308
This needs to be made more compact and user friendly.
Page 11-12 line 309-316
This does not read well and is somewhat confusing. The wording that cites research sounds like they are findings of the current research. Terminology is confusing. Is cannabis not a psychoactive substance. The point being made by the authors is unclear
Page 12 line 318-321
When comparisons are being made, it would be important to clarify if the other studies are reporting self-reports or toxicology results.
Page 12 line 323-355
I am not sure if this should be in the discussion. It may be better to focus on a core set of key findings giving policy implications.
Page 12 line 356-361
The policy implications of these findings need to be discussed. E.g. Does it suggest that social services may need to support and be involved with a larger proportion of mothers as there may be mothers with dependency issues that need support.
Page 12 line 362-366
Again policy implications need to be discussed.
Page 13 line 367-369
Again, is this a main discussion point
Page 13 line 374-379
The study limitations requires significant expansion.
Conclusions
Page 13 line 380-393
There is a need to include something about policy implications. For example is there a need to target initiatives at different social groups to help prevent drug use during pregnancy and provide support.
Author Response
- Reviewer 2
General
This is an important topic area and has important implications in terms of the future delivery of care to mothers and their new-borns, and also in terms of the use of toxicology results from meconium samples. However it does require significant improvements in terms of structure and focus, to ensure that key issues and policy implications can be determined. I hope the following detailed comments will facilitate the author(s) in revising the manuscript:
Simple summary
- Page 1 Line 35 Put percent in brackets after 49.
AMENDED. Percentage added.
Abstract
- Page 2 Line 47: Put percent in brackets after 49.
AMENDED. Percentage added.
- Page 2 Line 49-53: Put significance levels in for these results.
AMENDED. Significance levels have been added.
Introduction
- Page 2 line 63: A short overview of the key consequences for pregnant women and infants should be provided.
AMENDED. We added the sentence: Infants exposed to drugs of abuse during pregnancy have a greater risk of entering a Unit Neonatal intensive care and neonatal intermediate therapy. The impact of mothers substance use on their personal health and the health of their fetuses is a public health concern and…
- Page 2 line 68: “Allows identity maternal maternal,.,.”- suggest replace with “allows the identification of….”
AMENDED. We change the phrase.
- Page 2 line 71-72: “of drug detection window”- this sentence does not flow correctly and requires minor rewording.
AMENDED. We reworded the phrase: “…provide a long historical record of prenatal exposure to certain drugs and can account for different periods of gestation: meconium for the second and third trimester of gestation, fetal hair for the third”
- Page 2 line 83: It would be useful to highlight what percentage continue to use from other studies.
AMENDED. We added this information: “The cannabis use prevalence ranged from 8.1 % at the beginning of pregnancy to 2.5 in the last trimester (Alshaarawy & Anthony, 2019). Results from the 2013 National Survey on Drug Use and Health shoved that illicit drug was lower among pregnant women during the third trimester than during the first and second trimesters (2.4% vs. 9.0 and 4.8%). Similarly, alcohol use was lower during the second and third trimesters than during the first trimester (5.0 and 4.4% vs. 19.0 %)”.
In addiction it would be useful to explain the implications here of continued use during pregnancy.
A new paragraph has been added to explain these implications.
- Page 2 line 85: ‘drugs of abuse’- there needs to be clarity in terms of terminology throughout the paper. It should not be assumed that the reader knows what these drugs are. It is not clear whether the studies quoted refer to the general population or pregnant women, or neonatal infants etc
AMENDED. We added the principal drugs of abuse found in the studies. The studies quoted refer to newborns.
- Page 2 line 86-89: IT would be useful to provide background context in terms of SUD in Mallorca
AMENDED. We added information of SUD in Mallorca.
- Page 2 line 92: Expand stating why toxicology testing in pregnant women usually not recommended.
AMENDED. We have included a short overview to explain why toxicology testing in pregnant women usually is not recommended.
- Page 3 line 90-99: It might be worth starting with the Spanish position, and then state that similar positions are employed in other countries, with an absence use of toxicology data (giving reference to the American guidelines and also some from other countries.
AMENDED in accordance with the referee We change the paragraph.
- Page 3 line 98: ‘in the hands of’- this should be rephrased or put in single inverted commas.
AMENDED. We deleted the phrase.
- Page 3 line 101: Delete “In this scenario”
AMENDED.
- Page 3 line 124 Biomarker
AMENDED
Materials and Methods
- Page 3 line 134-136: I don’t understand why these were excluded. This exclusion requires clarification.
In general, the prenatal drug exposure studies published to date are carried out in maternity wards, where the proportion of preterm infants is very small or has been considered as an exclusion criterion. However, our study was carried out in a NICU, in which the admitted newborns have serious health problems and many of them are premature (40% of the cases included in the study were neonates with a gestational age of less than 34 weeks). This fact makes it difficult to obtain adequate meconium samples, especially in the clinical situations that have been considered as exclusion criteria in our study.
- Page 3 line 138: Short overview of meconium and rationale for use to analyse exposure, compared for example to hair or urine.
AMENDED. A short overview has been added.
- Page 4 line 141: What does EtG stand for?
AMENDED.
- Page 4 line 141-142: This is poorly explained. Why were these tests undertaken, and do they form part of the analysis in this paper?
AMENDED.
- Page 4 line 167: Further detail required. Was this before or after the baby was born? Was the written script a validated instrument, had it been piloted?
AMENDED. Mother interview was performed at admission of neonates in the NICU after the baby was born. The questionnaire was based on a questionnaires performed in previously reports. This information has been included in the manuscript.
- Page 4 line 174: Due to the fact that questions were asked about illegal substance use, I feel it would have been more appropriate to have used a self administered questionnaire. It would have been useful to correlate toxicology results with questions about psychoactive substances. The results appear to only show toxicology results. Is this correct and if this is the case, why were self reports collected?
The correlation of toxicology results with questions about psychoactive substances has been included in the results section.
- Page 4 line 179-189: Further information in terms of how the survey and toxicology data was matched should be given. Were procedures put in place to conform to GDPR. Were there any ethical issues that had to be addressed? Was it explained to participants that their survey data was to be matched with toxicology results.? Was permission required to undertake the toxicology tests? Further detail required which would benefit others considering similar research.
Local Human Research Ethics Committee approved study protocol was added (Research Ethics Committee of the Balearic Islands CEI-IB, project number IB 3538/17 IP).
- Page 4 line 187: I am not sure what was done here. Please explain/
A comparison of the sociodemographic profiles associated with the consumption of cannabis, cocaine and alcohol was carried out to detect differences between these profiles. Information in manuscript has been expanded to improve understanding.
- Page 5 line 198 ? 1.
AMENDED.
Results
- Page 5 line 203-206: A summary of the sociodemographic profile of participants should be provided.
AMENDED. A summary was added.
- Page 5 line 209 (13.2%)
AMENDED.
26.Page 5 line 211-216: Unless the authors provide justification, I feel that due to the small numbers involved, that this analysis is not required. However if the authors feel that this is important, then there will be a need to ensure the issues raised are included in the discussion
AMENDED. The paragraph has been deleted.
- Page 6-9 tables 1-2.: The tables are large and not particularly user friendly. The authors should revisit to determine whether there is a better way to present findings. For example consideration should be given to presenting the core findings as opposed to detailed disaggregation of a small sample.
We have followed your recommendation to make the tables simpler and user-friendly. The results of tables 1 and 2 have been combined into a single table (new Table 1) that shows only those variables with statistically significant differences between groups.
- Page 10 line 234-253: These results are presented in a laborious and repetitive way. The key results should be presented in the text in a summarised user friendly format, with detailed results provided in table format. Currently too many patterns and relationships are presented which can mask the key patterns emerging. In addition, the numbers involved for some of the patterns are small, and I would query whether the size is sufficient to permit such analysis.
Section 3.2 has been rewritten and summarized with reference to the new table 1, which, as indicated, has also been redone from tables 1 and 2.
- Page 10 line 279: How is “lack of pregnancy care” defined.
AMENDED. The lack of pregnancy care was defined in discussion section (line 343).
Discussion
- Overall the discussion lacks focus. It presents lots of relationships that were found and then cites similar findings from the literature. Whilst comparisons withother studies is generally useful, in this case it fails to provide any real impact in terms of core issues of concern. There are key findings in the study, but these are lost in the patterns and relationships that are reported. I feel that a more focused discussion would have been more appropriate, perhaps structuring around the core drivers of the multiple logistical regression output. It would be better to take each of these, compare to other studies and discuss the implications. The utility of using meconium for toxicology results also warrants discussion.
Discussion has been modified and rewritten, including the utility of using meconium for toxicology results.
- Page 11 line 297: NICU?
NICU has been previously defined in the Introduction.
- Page 11 line 301-308. This needs to be made more compact and user friendly.
AMENDED.
- Page 11-12 line 309-316. This does not read well and is somewhat confusing. The wording that cites research sounds like they are findings of the current research. Terminology is confusing. Is cannabis not a psychoactive substance. The point being made by the authors is unclear
AMENDED. The paragraph has been rewritten in order to clarify its meaning.
- Page 12 line 318-321: When comparisons are being made, it would be important to clarify if the other studies are reporting self-reports or toxicology results.
As in our work, in all the studies mentioned tobacco consumption was declared by the mother and no toxicological analyses were performed. This information was included in the manuscript.
- Page 12 line 323-355: I am not sure if this should be in the discussion. It may be better to focus on a core set of key findings giving policy implications.
Following your suggestion, these paragraphs have been shortened substantially to focus on the key findings that give policy implications.
- Page 12 line 356-361: The policy implications of these findings need to be discussed. E.g. Does it suggest that social services may need to support and be involved with a larger proportion of mothers as there may be mothers with dependency issues that need support.
Policy implications of these findings have been briefly discussed.
- Page 12 line 362-366: Again policy implications need to be discussed.
Policy implications of these findings have been briefly discussed.
- Page 13 line 367-369: Again, is this a main discussion point
Policy implications of these findings have been briefly discussed.
- Page 13 line 374-379: The study limitations requires significant expansion.
AMENDED.
Conclusions
- Page 13 line 380-393: There is a need to include something about policy implications. For example is there a need to target initiatives at different social groups to help prevent drug use during pregnancy and provide There is a need to include something about policy implications
Some aspects of policy implications have been included in the Conclusions
